



# Response to Filchner-Ronne Ice Shelf cavity warming in a coupled ocean—ice sheet model. Part I: The ocean perspective

Ralph Timmermann[1] and Sebastian Goeller[1,2]

[1]Alfred-Wegener-Institut Helmholtz-Zentrum für Polar- und Meeresforschung, Bremerhaven, Germany
[2]now at GFZ German Research Centre for Geosciences, Potsdam, Germany
*Correspondence to:* Ralph Timmermann (Ralph.Timmermann@awi.de)

**Abstract.** A Regional Antarctic and Global Ocean (RAnGO) model has been developed to study the interaction between the world ocean and the Antarctic ice sheet. The coupled model is based on a global implementation of the Finite Element Sea-ice Ocean Model (FESOM) with a mesh refinement in the Southern Ocean, particularly in its marginal seas and in the sub-ice shelf cavities. The cryosphere is represented by a regional setup of the ice flow model RIMBAY comprising the Filchner-Ronne Ice Shelf and the grounded ice in its catchment area up to the ice divides. At the base of the RIMBAY ice shelf, melt rates from FESOM's ice-shelf component are supplied. RIMBAY returns ice thickness and the position of the grounding line. The ocean model uses a pre-computed mesh to allow for an easy adjustment of the model domain to a varying cavity geometry.

RAnGO simulations with a 20th-century climate forcing yield realistic basal melt rates and a quasi-stable grounding line position close to the presently observed state. In a centennial-scale warm-water-inflow scenario, the model suggests a substantial thinning of the ice shelf and a local retreat of the grounding line. The potentially negative feedback from ice-shelf thinning through a rising in-situ freezing temperature is more than outweighed by the increasing water column thickness in the deepest parts of the cavity. Compared to a control simulation with fixed ice-shelf geometry, the coupled model thus yields a slightly stronger increase of ice-shelf basal melt rates.

## 1 Introduction

Mass flux from the Antarctic ice sheet to the Southern Ocean is dominated by iceberg calving and ice-shelf basal melting. Until recently, it was assumed that iceberg calving was the dominant sink of Antarctic ice sheet mass, but ice-shelf basal melting is now estimated to outweigh any other process (Depoorter et al., 2013; Rignot et al., 2013). Ice shelves have been shown to buttress the flow of outlet glaciers and ice streams (e.g., De Angelis and Skvarca, 2003; Dupont and Alley, 2005). Changes in ice-shelf thickness and grounding location therefore alter the discharge of grounded ice, which then contributes to global sea level rise. The acceleration of mass loss from the Antarctic ice sheet since the 1990s (Rignot et al., 2011) has been attributed to enhanced ice-shelf basal melting and related ice-shelf thinning particularly in the Amundsen/Bellingshausen Seas sector (Pritchard et al., 2012).

Models of ice shelf–ocean interaction are not only used in hindcasts or sensitivity studies, but also in attempts to project future melt rates, either with idealized changes in forcing (e.g. Kusahara and Hasumi, 2013) or with atmospheric forcing



derived from coupled climate model projections. Using atmospheric output from the HadCM3 climate model, Hellmer et al. (2012) found the potential of a rapid warming of the southwestern Weddell Sea continental shelf waters by a redirected coastal current. In the Jacobs et al. (1992) terminology, the replacement of cold shelf water by water originating from Warm Deep Water (WDW) leads to a shift from Mode 1 to Mode 2 melting and thus to dramatially increased melt rates for Filchner-Ronne

Ice Shelf (FRIS). Timmermann and Hellmer (2013) showed that the surface freshwater flux on the Weddell Sea continental shelf, which is governed by sea ice formation and thus largely determined by atmospheric forcing, is critical in allowing or preventing this transition in the melting mode. Observational evidence of warm pulses already arriving at the ice-shelf front (Darelius et al., 2016) indicates that this is a realistic scenario.

All model studies mentioned above assumed a static ice-shelf geometry even with simulated melt rates near the grounding

line rising to almost 20 m/yr (Timmermann and Hellmer, 2013). To overcome this deficiency and study ice shelf—ocean interaction in a warming climate in a consistent way, we coupled FESOM to a regional setup of the ice flow model RIMBAY (Thoma et al., 2014) and forced the coupled model with output from HadCM3 that has been obtained for present-day climate and the A1B scenario (Collins et al., 2011). This paper describes the coupling procedure and reports on the solutions we found for a suite of technical challenges (Chapter 2). A 250-years long coupled model run with climate-projection forcing serves as

the reference simulation and is compared to control experiments with (i) continuous present-day forcing and (ii) static ice-shelf geometry (Chapter 3). The focus of the analysis here is on processes and sensitivies in the sub-ice shelf cavity. Ice dynamics and ice sheet mass balance will be discussed in the corresponding paper Part II: the ice perspective (Goeller and Timmermann, 2017).

## 2   Regional Antarctic ice and Global Ocean Model (RAnGO)

### 2.1   Overview

RAnGO combines a regional model of the Antarctic ice sheet with a global ocean model. The coupled system consists of a global configuration of the Finite Element Sea-ice Ocean Model (FESOM; Timmermann et al., 2012), and a regional setup of the Revised Ice Model Based on frAnk pattYn (RIMBAY; Thoma et al., 2014). While the FESOM domain covers the world ocean including the sub-ice shelf cavities in the Southern Ocean, the RIMBAY setup comprises the Filchner-Ronne Ice Shelf

(FRIS) and the relevant catchment basin up to the ice divides. The interface between the two models is the FRIS base (Fig. 1). As for the standalone model runs of Hellmer et al. (2012), the coupled model is forced by atmospheric output from the HadCM3 climate model.

### 2.2   The ocean component: FESOM

FESOM is a primitive-equation hydrostatic ocean model that is solved on a horizontally unstructured mesh (Wang et al., 2014).

It comprises a dynamic-thermodynamic sea ice model (Danilov et al., 2015). The ice-shelf component (Timmermann et al.,





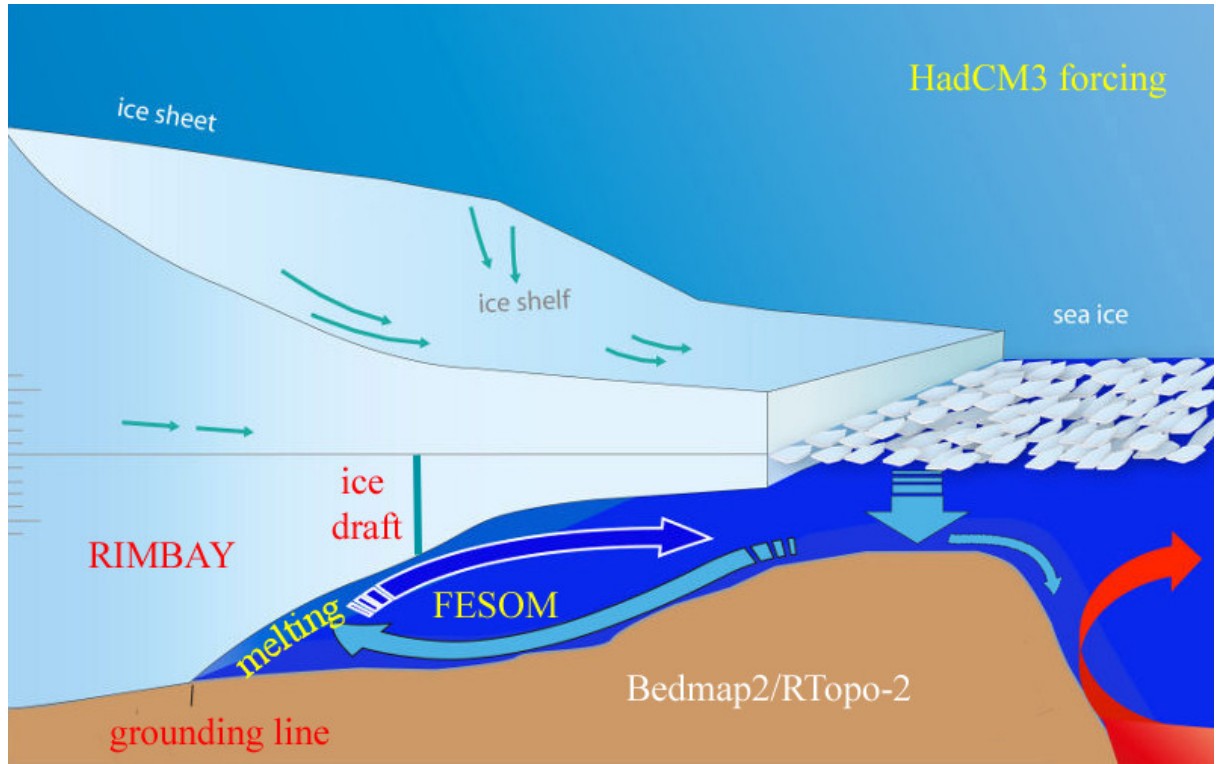

**Figure 1.** Schematic representation of the Filchner-Ronne Ice Shelf cavity, including cavity / shelf water circulation for the present-day Mode 1 melting / cold-water ice shelf scenario.

2012) goes back to the Hellmer and Olbers (1989) 3-equation model of ice shelf—ocean interaction with a velocity-dependent parameterization of boundary layer heat and salt fluxes according to Holland and Jenkins (1999).

The model is run on a global mesh with a horizontal resolution varying from 1 km along the FRIS grounding line to 340 km in the deep Atlantic and Pacific basins (Fig. 2). Is uses a hybrid vertical coordinate with 22 sigma levels south of the 2500-m isobath surrounding the Antarctic continent and up to 36 z-levels outside this domain. Antarctic sub-ice shelf cavities are thus all inside the sigma-domain, which allows for a smooth representation of the ice-shelf base. The ice-shelf front is approximated in a ramp-like shape; with a horizontal resolution between 10 and 16 km in this area the deviation from the true geometry is confined to 50 (100) km inwards of the Filchner (Ronne) ice-shelf front. We apply a minimum water column thickness of 50 m for all sub-ice cavities.

An early version of RTopo-2 (Schaffer et al., 2016) has been used to derive ocean bathymetry and the ice-shelf draft and grounding lines for all cavities with fixed geometry. With present-day ice-shelf configuration for FRIS, the FESOM mesh comprises a total of $\approx 2.6 \times 10^6$ nodes, $1.1 \times 10^5$ of which are surface nodes (where the term *surface* equally refers to open ocean and ice-shelf base). The model is run with a default time step of 90 seconds owing to the very fine horizontal resolution



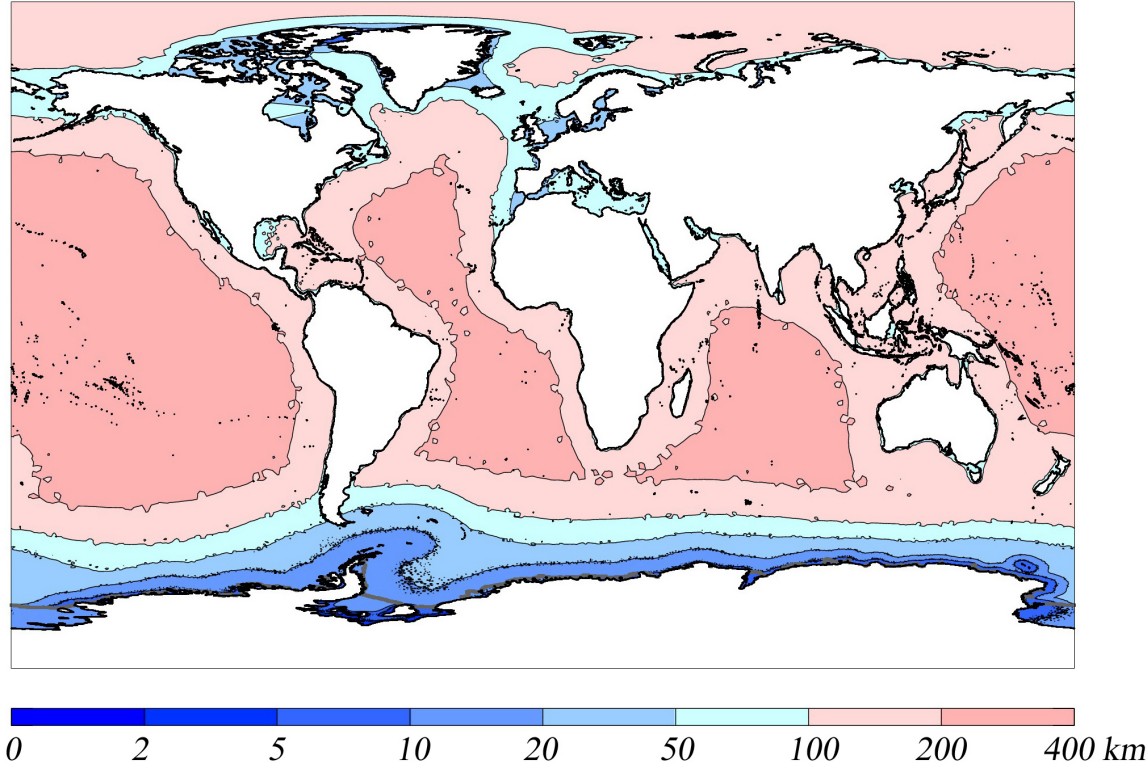

**Figure 2.** Horizontal resolution of RAnGO's ocean component. Note the nonlinear color scale.

along the FRIS grounding line and a minimum sigma layer thickness of just over 2 m. For several situations with eddies running into shallow sections of the FRIS cavity, it turned out to be necessary to decrease the ocean model time step to 4 seconds.

### 2.3 The ice component: RIMBAY

RIMBAY (Thoma et al., 2014) is a threedimensional thermomechanical multi-approximation ice shelf / sheet model going
5   back to the ice flow model of Pattyn (2003). Within the RAnGO experiments, the ice model domain comprises the FRIS and its upstream catchment area of grounded ice, confined by the surrounding ice divides (Fig. 3). Ice velocities are calculated following the hybrid approach which combines the shallow-ice and the shallow-shelf approximations. A basal friction correction at the grounding line after Feldmann et al. (2014) ensures a smooth transition between grounded and floating ice and thus a realistic grounding line migration.

10   The ice model is run at a horizontal resolution of 10 km with 41 terrain-following sigma layers and a time step of 0.1 yr. Bedmap-2 (Fretwell et al., 2013) data are used for bedrock topography and initial ice thickness. Since Bedmap-2 is also the source for the Antarctic ice and bedrock relief in RTopo-2, all topographies are fully consistent within RAnGO.



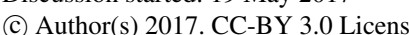

**Figure 3.** Antarctic ice sheet with coast and grounding lines (black) and the ice divides (green and blue) from Antarctica's Gamburtsev Province Project (AGAP). The green lines indicate the RIMBAY model domain in this study.

## 2.4 Model spin-up and coupling

First, we perform a 1000-yr stand-alone RIMBAY simulation. This ice model spin-up is forced by present-day surface temperatures (Comiso, 2000), accumulation rates (Arthern et al., 2006) and geothermal heat flux (Shapiro and Ritzwoller, 2004). Basal melt rates are parameterized according to Beckmann and Goose (2003). As a result, ice dynamics are in a quasi-stationary
5  steady state and ice thickness, ice velocity and grounding line position match current observations very well. Additional figures and a thorough discussion of the ice model spin-up are presented in Part II: the ice perspective (Goeller and Timmermann, 2017).

With this *RIMBAY present-day* cavity geometry, we integrate FESOM for 21 years (1930–1950) using atmospheric forcing from the 20th-century simulation of the HadCM3 climate model. Annual mean basal melt rates for FRIS from the last year
10  (1950) are then transfered back to RIMBAY, which starts the RAnGO coupled model loop (Fig. 4).

For each of the cycles within the coupled RAnGO system, FRIS basal melt rates averaged over year $N$ are obtained from FESOM and passed to RIMBAY, which is then stepped forward for that same year $N$. From the simulated ice draft and grounding line location at the end of RIMBAY year $N$, a new cavity geometry is derived and an updated FESOM mesh is generated. FESOM's prognostic variables are projected onto the new mesh (details below), and FESOM is then integrated over
15  year $N + 1$. Basal melt rates are averaged over FESOM year $N + 1$ and passed to RIMBAY, and the cycle repeats itself with RIMBAY running year $N + 1$.

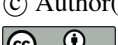



## Launch procedure and coupling

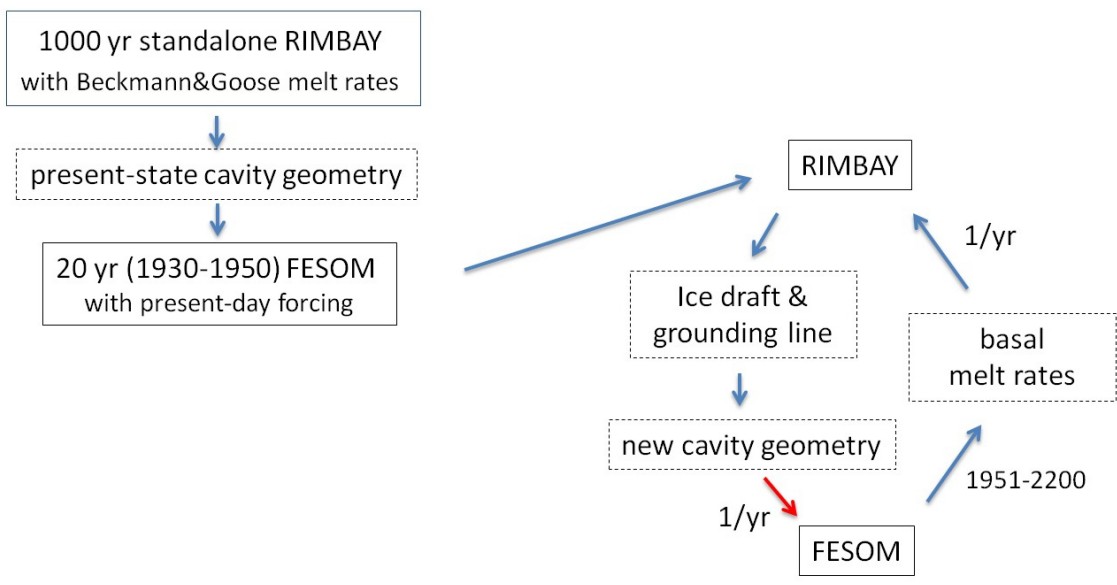

**Figure 4.** RAnGO initialization and coupling scheme. The red arrow indicates the most time-consuming element of the model coupling.

A one-year coupling time step is long compared to the relevant time scales in the ocean but short compared to the typical time scales for ice dynamics or the ice mass budget. The fact that variations of RIMBAY ice thickness distribution, grounding line location, and melt rate patterns are small for each coupling time step indicates that this coupling strategy is adequate.

### 2.5 Dynamic FESOM mesh modification

Adjustment of the model domain to a varying ice-shelf geometry is a natural part of RIMBAY and is rather straightforward to implement for a finite-difference ocean model with a land-sea mask. For FESOM, the computational mesh only exists in the ocean and has to satisfy certain criteria in order to ensure numeric stability and efficiency. For standalone FESOM applications (e.g., Timmermann et al., 2012) we use an iterative method to generate surface meshes in which the size of triangles smoothly varies according to the desired resolution, while at the same time triangles approximate the ideal of being equilateral as good

as possible. For a mesh with about $10^5$ surface nodes, the algorithm takes about 2 days to converge - which clearly prevents it from being used as part of a coupling interface.

To reduce the mesh generation overhead, we generated an initial surface grid that covers the full RTopo-2 ocean (blue and green triangles in Fig. 5) plus all ice around FRIS that is grounded on bedrock deeper than 100 m below sea level (red triangles in Fig. 5). This criterion to define the "potentially-ungrounding" area adjacent to the currently floating ice shelf proves to be

well on the safe side for any grounding line movement in our coupled model runs. Mesh resolution along the present-day grounding line and in the potentially-ungrounding area is about 1 km to describe grounding line migration as smoothly as



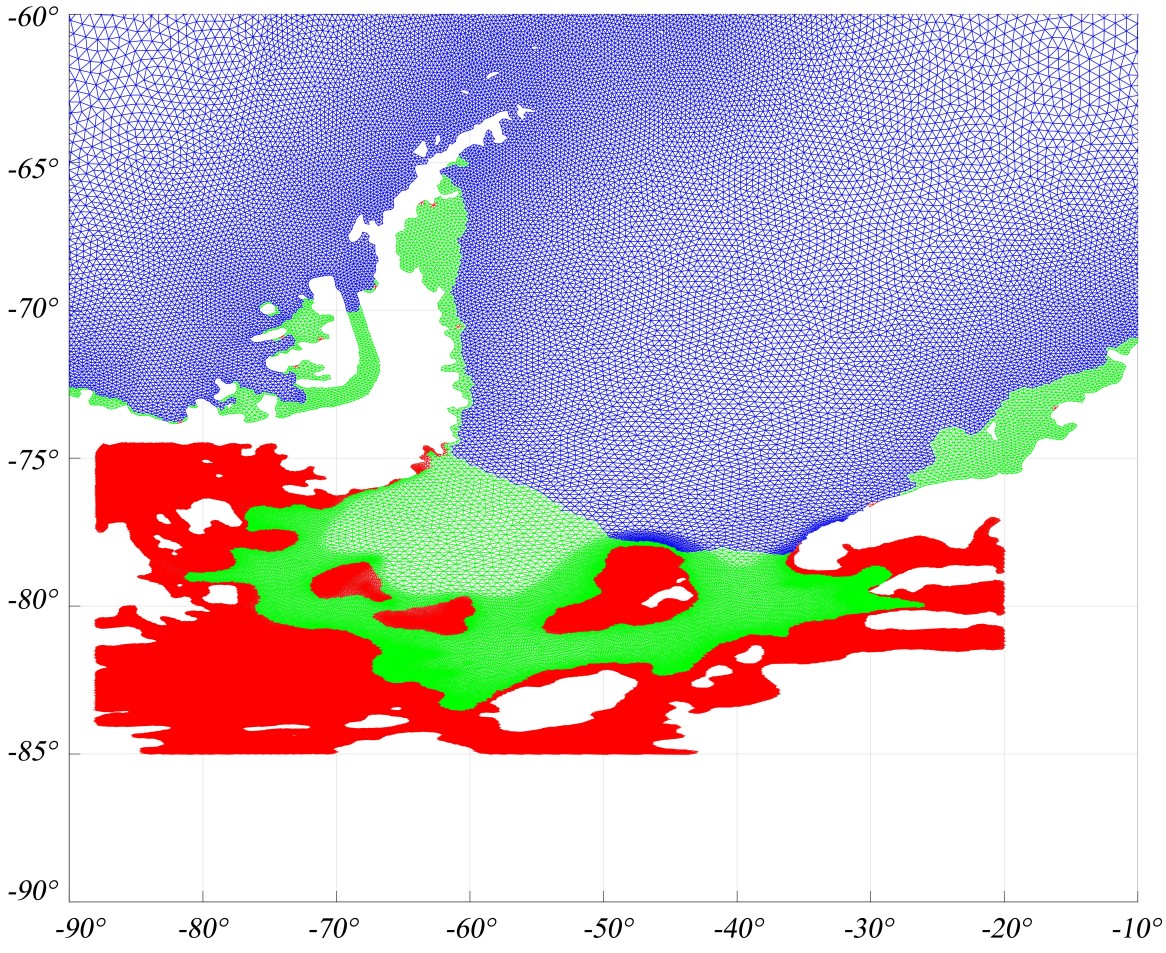

**Figure 5.** Precomputed and actual FESOM mesh in the Weddell Sea / FRIS sector: Blue triangles indicate open ocean, green triangles indicate ice-shelf cavities in the RAnGO geometry for simulated year 2000 (i.e. at the end of the 20th-century spinup). Red triangles refer to elements that have been created in the initial mesh but are removed for the year 2000 geometry because they were found to be covered with grounded ice (potential ocean mesh in a warming scenario).

possible. For each topography/cavity geometry to be run with RAnGO, the coupler removes all grid nodes that are covered by grounded ice. The remaining ocean grid nodes are renumbered consecutively. The full three-dimensional grid is then created from this new surface mesh, with the terrain-following vertical coordinate easily adjusting to any change in water column thickness due to a varying ice-shelf draft.

5      During this procedure, the vast majority of finite element mesh nodes keep their position (despite being renumbered), so that no horizontal interpolation is necessary for the ocean state variables outside the immediate vicinity of the grounding line. This makes it much easier to ensure the conservation of heat and salt. Wherever new ocean (i.e. cavity) nodes are created, ocean



temperature, salinity, and sea surface height are taken from the nearest existing neighbor grid node. Again, the small variations per coupling step make this simple "no-flux" approach justified.

## 2.6   Computational load

Coupling has been implemented in an "offline" way with RIMBAY and the RAnGO coupler running on local servers and
FESOM relying on a massively parallel supercomputing system. Given that (1) model output is in any case transfered to local disks for postprocessing and analysis and that (2) the updated mesh configuration files are comparatively small, the overhead arising from the file transfer necessary in our offline approach is negligible.

   To run a typical model year with the current configuration of RAnGO requires about 7 hours on 528 CPUs for FESOM, less than 10 minutes for RIMBAY, and almost 2.5 hours for the coupling procedures. Within the coupler, more than 90% of the
time are spent on the construction of the threedimensional, tetrahedral mesh from the updated surface grid. A more efficient algorithm that starts from the existing threedimensional mesh and only applies corrections where necessary is currently being developed.

## 2.7   Experiments

As stated in Section 2.4, 1951 is the first year of the coupled RAnGO simulation. We integrated the coupled model until 1999
using atmospheric output (10-m wind speed, 1.5-m air temperature, 1.5-m specific humidity, surface moisture flux, downward long- and shortwave radiation, total precipitation) from the HadCM3 20th-century simulation. This experiment is refered to as the *RAnGO 20C* simulation. HadCM3 data for the A1B scenario have been used to conduct the *RAnGO A1B* simulation for the period 2000-2199. The suite of the *RAnGO 20C (1950-1999) + A1B (2000-2199)* model runs serves as the reference simulation for analysis in this paper. A control run with present-day climate (*RAnGO CTRL*) has been performed twice repeating the
HadCM3 1900-1999 forcing.

   Next to the coupled RAnGO simulation launched from the end of 1950, an uncoupled FESOM experiment with the *RIMBAY present-day* cavity geometry prescribed has been conducted. Like its RAnGO counterpart, this experiment starts with a *FESOM 20C* simulation and splits into a *FESOM A1B* and a *FESOM CTRL* branch at the beginning of the 21st century.

## 3   Results

### 3.1   Ice-shelf basal melt rates and hydrography

### 3.1.1   FESOM and RAnGO 20C experiments

Time series of simulated mean basal melt rates in the *FESOM 20C* and *RAnGO 20C* simulations (black and blue lines in Fig. 6) indicate that ice shelf—ocean interaction approaches a quasi-steady state within less than a decade after the initialization with a relatively warm water mass in the FRIS cavity. Mean basal mass loss over the period 1950-1999 amounts to 87 Gt/yr in the
fixed-geometry FESOM experiment and to 93 Gt/yr in the fully coupled RAnGO model run. Both are well within the range





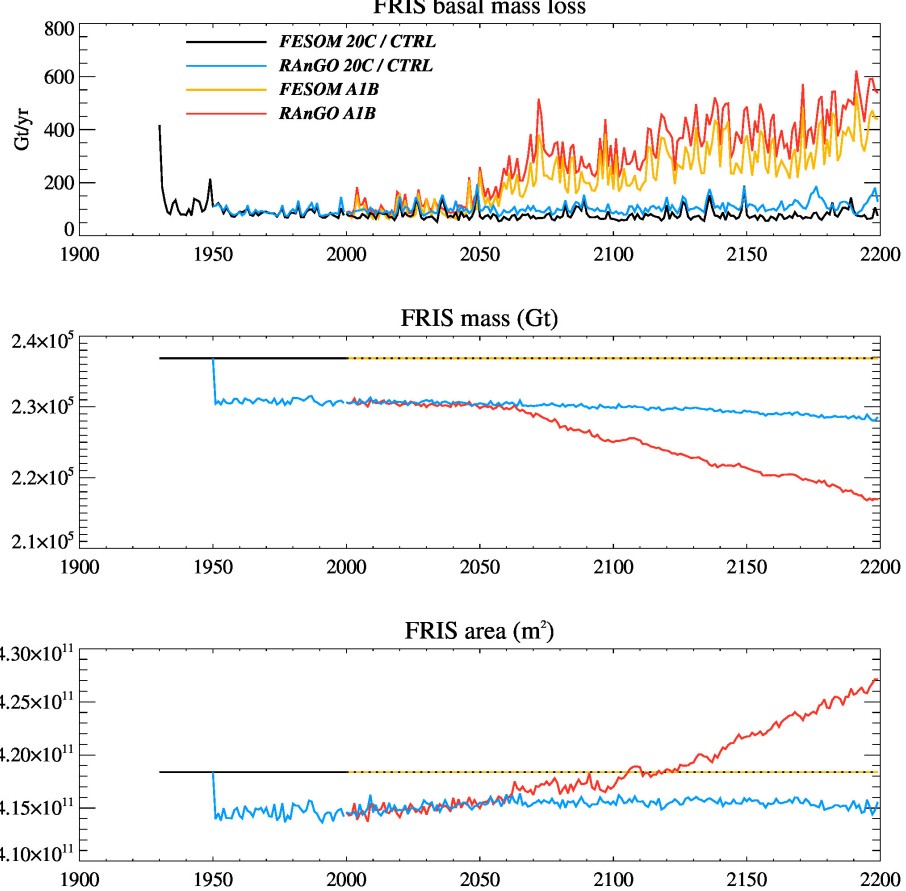

**Figure 6.** Time series of annual-mean basal melt rate, ice-shelf mass, and ice-shelf area for FRIS in fixed-geometry FESOM experiments with 20th century (black line) and A1B (yellow line) forcing and in RAnGO experiments for the 20th century (blue line) and the A1B scenario (red line).

of observational estimates (e.g., $50 \pm 40$ Gt/yr [Depoorter et al., 2013] vs. $155 \pm 35$ Gt/yr [Rignot et al., 2013]); the difference between the two experiments is much smaller than the (modelled) interannual variability.

Maximum melt rates for present-day climate are about 5 m/yr and occur in the deepest parts of the cavity close to the grounding lines of Support Force Glacier and Foundation Ice Stream (see Fig. 9 for locations), where the ice base reaches 1100 and 1400 m below sea level, respectively, and the *in-situ* freezing point is about 1 K below the surface freezing point (Fig. 7, left panels). Melt rates between 3 and 5 m/yr are suggested for Evans and Rutford Ice Streams in the western sector of Ronne Ice Shelf, which is consistent with estimates based on ice flux divergence (Joughin and Padman, 2003).



**Figure 7.** Top: 10-yr mean basal melt rates in the *RAnGO 20C* experiment for 1990-1999 (left), in *RAnGO A1B* for 2190-2199 (middle), and in *FESOM A1B* for 2190-2199 (right). Bottom: Corresponding ice-shelf drafts from the *RAnGO 20C/A1B* simulation for 1995 (left) and 2195 (middle), and in the *RIMBAY present-day* geometry. Coloured areas represent modelled cavity geometries. Black lines denote coast and grounding lines from RTopo-2.

An extended area of marine ice formation is suggested north of (i.e. downstream from) the Henry and Korff Ice Rises. While this pattern is consistent with the observed locations of marine ice (Lambrecht et al., 2007), accretion rates in most of the refreezing area are smaller than in the estimates of Joughin and Padman (2003).

Two additional hot spots of marine ice formation (at rates exceeding 1.0 m/yr) are associated with the outflow of Ice Shelf
5 Water (ISW) in the Filchner and Ronne Troughs. For Filchner Trough, this is again consistent with Joughin and Padman (2003), although their data and the marine ice thicknesses observed by Lambrecht et al. (2007) suggest a more pronouned





freezing pattern on the western side. For the western side of Ronne Trough, marine ice formation along the coast is consistent with the sub-ice circulation suggested by Nicholls et al. (2004).

Modelled present-day cavity geometry agrees well with the location of grounding lines in Bedmap2, except for the three narrow ice streams feeding the western sector of Ronne Ice Shelf. Due to small ice thickness and bedrock gradients, the grounding line positions in this area are highly sensitive to ice thickness changes during the RIMBAY spin-up. A stringent validation of modeled FRIS topography is provided in Part II of this paper.

### 3.1.2 FESOM and RAnGO A1B projections

With the beginning of the 21st century, but most notably after 2050, the two experiments start to deviate from each other according to the scenario chosen (see Fig. 6). The A1B simulation with RAnGO (and also with fixed-geometry FESOM) features a rapid rise of FRIS basal melt rates during the second half of the 21st century, followed my a more gradual increase during the 22nd century. By the 2190s, basal mass loss for FRIS has increased to about 540 Gt/yr in the *RAnGO A1B* simulation, which corresponds to a factor of six compared to the 20th-century situation. Melt rates along the grounding line in this situation (Fig. 7, top middle panel) exceed 12 m/yr; areas of refreezing have vanished almost completely. In contrast to the 1990s case, there now is a strong signature of Jacobs et al. (1992) "Mode 3" melting with melt rates up to 20 m/yr along the Filchner ice front.

The strongest increase in basal melt rates occurs between 2050 and 2070. Like in the experiments of Hellmer et al. (2012) and Timmermann and Hellmer (2013), this is caused by a flow of Modified Warm Deep Water (MWDW) onto the continental shelf and into the sub-ice shelf cavity (Fig. 8). In contrast to the former experiments, a water column thickness of only 120 m (90 m) southwest of Henry Ice Rise prevents the warm water from flushing even larger parts of the Ronne cavity in the RAnGO (FESOM) simulations discussed here.

The change in hydrography in the second half of the 21st century is very similar between the coupled RAnGO simulation and the uncoupled FESOM experiment with fixed cavity geometry (not shown). Area-mean basal melt rates in *FESOM A1B* follow the *RAnGO A1B* evolution very closely until about 2050 (top panel in Fig. 6). Differences increase within a few decades after the onset of cavity warming, with the fixed-geometry melt rates always staying below their RAnGO counterparts. With 418 Gt/yr for *FESOM A1B*, the mean basal mass loss over the period 2190-2199 is about 20 % lower than in the coupled RAnGO simulation. The distribution of melt rates, however, is very similar between *RAnGO A1B* and *FESOM A1B* (Fig. 7, top middle and right panels).

Throughout the integration, the interannual (year-to-year) variability of area-mean melt rates is very similar between the coupled and the uncoupled model runs. Summer-intensified "Mode 3" melting along the ice front is the dominant mechanism here; its year-to-year fluctuations are governed by variations of summer sea ice extent / summer ocean surface heating. The magnitude of these anomalies and the relative importance of "Mode 3" melting increase during the 21st and 22nd centuries as a response to a decreasing sea ice cover in the southern Weddell Sea: With increasing areas of open water, sea surface temperature can diverge from the surface freezing point temperature more easily, so that interannual variability leaves a stronger footprint on the properties of water getting in touch with the ice-shelf base along the ice front.

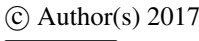



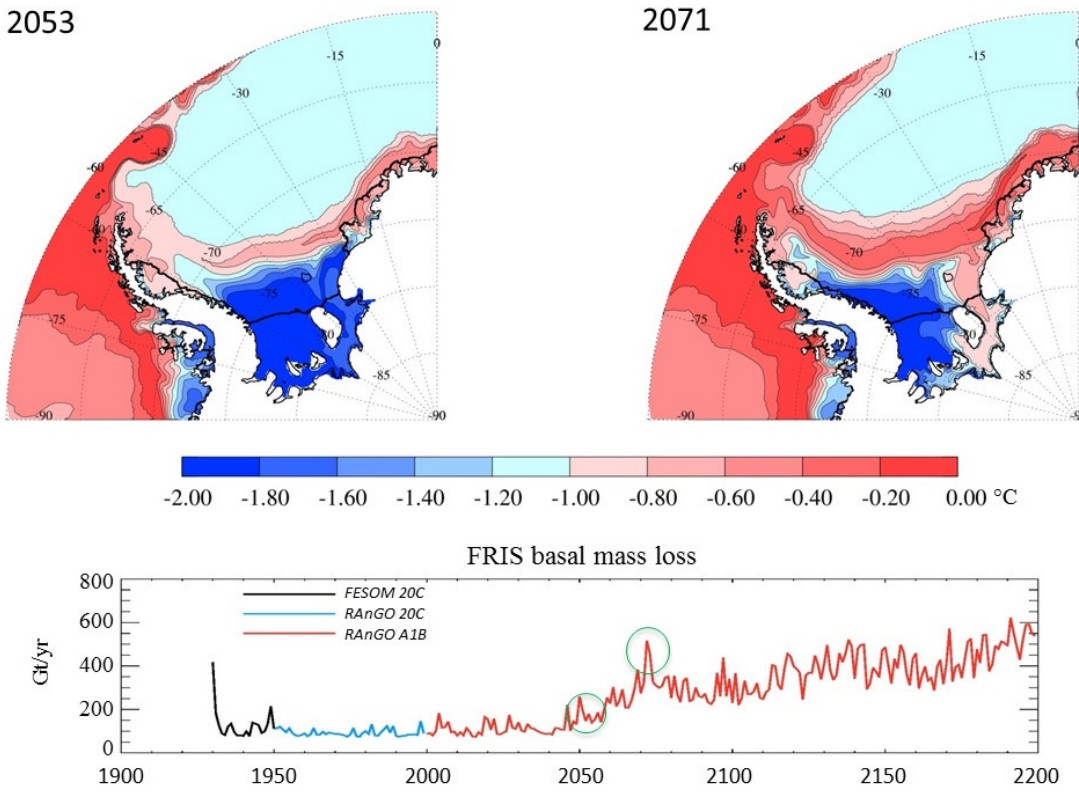

**Figure 8.** Top: Simulated bottom temperature for 2053 and 2071 in the *RAnGO A1B* experiment. The lower panel indicates the corresponding points in time on the time series of annual-mean basal mass loss.

### 3.1.3 Present-day climate control experiments

In contrast to the regime shift suggested by the *A1B* experiments, the control runs with a perpetual 20th-century forcing (black and blue lines in Fig. 6) preserve the "cold-water ice shelf" state of the original *FESOM 20C* and *RAnGO 20C* simulations with only little change in the distribution and area-average of basal melt rates. The rapid cavity warming caused by a inflow of
5   MWDW does not occur in these simulations.

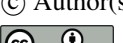



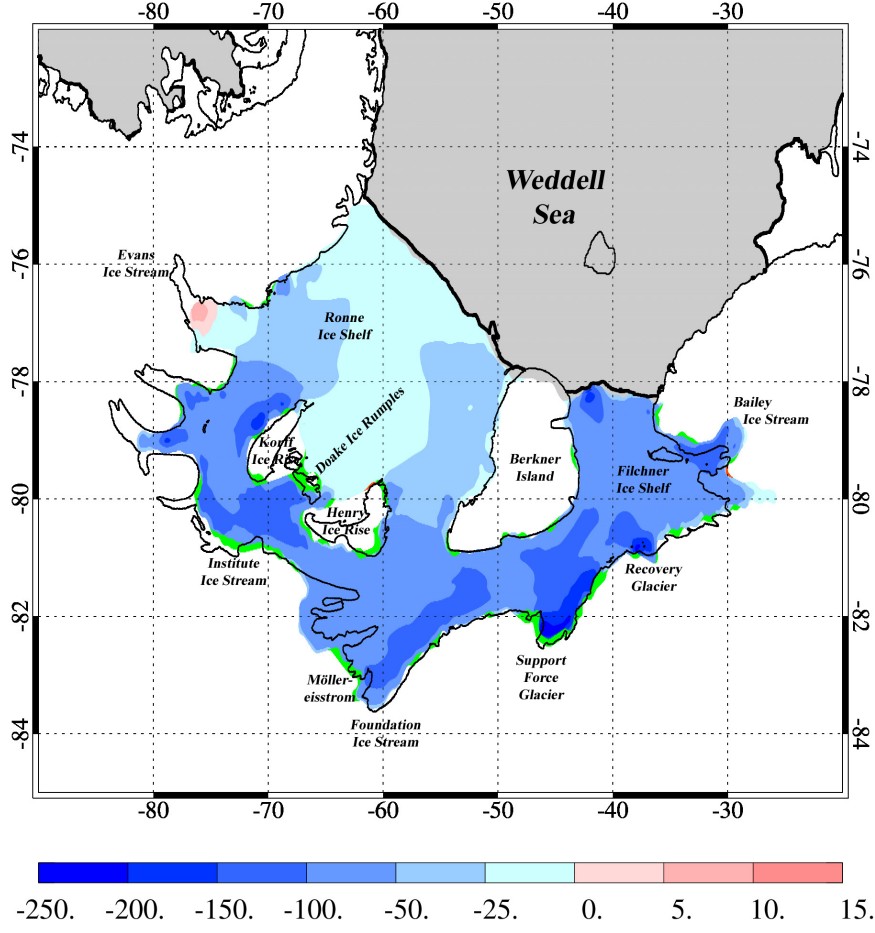

**Figure 9.** Simulated ice draft change (m) for FRIS from 1995 to 2195 in the *RAnGO 20C/A1B* experiment. Note the nonlinear color scale. Green color indicates areas of originally grounded ice that becomes afloat. Two small red patches represent areas of floating ice that becomes grounded. Thin (thick) black lines indicate coast/grounding lines (ice shelf fronts) from RTopo-2.

## 3.2 Ice shelf thickness, area and mass

### 3.2.1 RAnGO A1B experiments

Owing to the increasing basal melt rates in the A1B scenario, FRIS in the *RAnGO A1B* simulation continuously loses mass from 2050 onwards (Fig. 6, middle panel). Between the 1990s and the 2190s, FRIS mass decreases by $1.4 \times 10^4$ Gt (i.e. 6.1 %).

5  Consistent with the location of the highest melt rates, the strongest thickness decrease occurs at the inflow of Support Force Glacier (Fig. 9). At the maximum, ice-shelf draft (thickness) is reduced by 225 (286) m between 1995 and 2195 here. This is



also one of the few locations with a substantial grounding line retreat (green areas in Fig. 9). Despite the fact that melt rates (and the melt rate increase) are of similar magnitude at the floating parts of Foundation Ice Stream, ice draft reduction does not exceed 150 m and the grounding line remains stable there. This discrepancy will be analysed in Part II of this paper.

With the grounding line retreating also at the Möllereisstrom, the Institute Ice Stream, and at the Henry and Korff Ice Rises, so that the Doake Ice Rumples become detached from the ice-shelf base, the area of floating ice (Fig. 6, lower panel) increases by $1.15 \times 10^{10}\,\mathrm{m}^2$ (i.e. 2.8 %).

While ice-shelf thickness trends are small for the northern sector of Ronne Ice Shelf, a substantial thinning is also found at the Filchner Ice Shelf front. This is associated with the increasing rate of "Mode 3" melting during the increasingly long summer season (top right panel in Fig. 7) and thus reflects the impact of a decreasing sea ice coverage / increasing summer sea surface temperature on basal melting near the ice-shelf front.

### 3.2.2 Present-day climate control experiment

In the RAnGO control run with perpetual 20th-century forcing, FRIS mass at the end of the 22nd century differs from the 1990s state by less than 1 % (blue line in Fig. 6, middle panel). In the first half of the 21st century, a positive trend in the floating ice area is very similar between the control run and the A1B experiment (blue and red lines in Fig. 6, bottom panel), which indicates that the grounding line location for simulated FRIS in RAnGO is not in a strict steady state even for perpetual 20th-century forcing. Given the indications for a gradual FRIS thickness loss during the first decade of the 21nd century (Paolo et al., 2015) this might well be consistent with the true current situation. In any case, this model background variability is very small compared to the divergence between the two simulations after ≈ 2070.

### 3.3 Thickness—melt rate feedback

As has been discussed in Section 3.1.2, simulated basal melt rates increase by roughly a factor of six between the 1990s and the 2190s in the *RAnGO A1B* and *FESOM A1B* simulations. Taking a more quantitative view, we find an increase factor of 6.1 for *RAnGO A1B* vs. only 5.4 for *FESOM A1B*. We also note that the area-mean melt rate in *RAnGO A1B* is always higher than in *FESOM A1B* (Fig. 6). Given that a reducing ice draft implies a rising *in-situ* freezing temperature and thus a reducing melt potential for any warm water mass flowing into the cavity, this is not necessarily an obvious result - instead it would have appeared plausible to assume that a reducing ice thickness would reduce (not increase) the basal melt rate in a warming scenario like the one discussed in this study. In this section, we will therefore look into the processes that lead to an increased melt for a thinning ice shelf.

From the RAnGO and FESOM melt rate maps for the 2190s (Fig. 7 top middle and right panels) a systematic change introduced by the transition from a fixed-geometry to a coupled model is not obvious. A map of the difference between the two fields (Fig. 10, left panel) reveals that in many places with a retreating grounding line, an increased melt in newly ungrounded areas is at least partly compensated by reduced melt in areas along the old grounding line location. Even in this warm-water-inflow scenario, ice shelf–ocean interaction in many places still appears to fully extract the heat content of water getting in





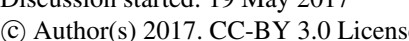

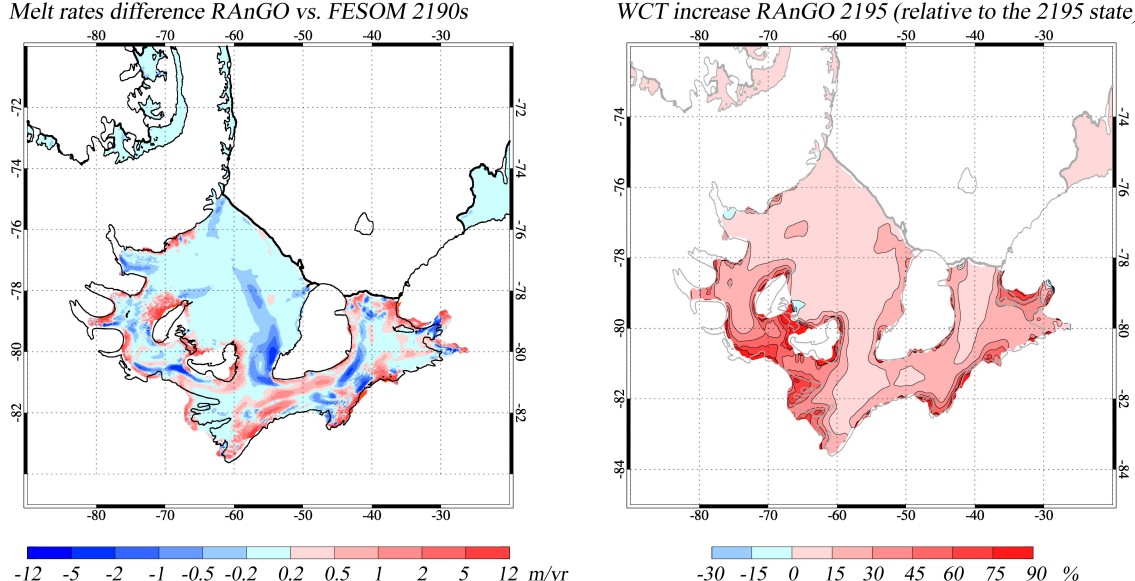

**Figure 10.** Left: Difference in simulated mean melt rates for the period 2190-2199 in coupled and fixed-geometry simulations (RAnGO minus FESOM). Right: Increase of water column thickness (wct) from *FESOM A1B* to *RAnGO A1B* relative to the *RAnGO A1B* 2195 state. For example, an increase of 90% at a given location means that 90% of the water column thickness found in 2195 have been created by ice-shelf thinning.

touch with the ice base near the grounding line, so that a shift in the grounding line position merely shifts the location of melt rate maxima, but does not increase total melt.

A more substantial and at least partly unbalanced melt rate increase is found (1) along the ice-shelf front and (2) at locations with a substantial increase of water column thickness. Along the ice-shelf front, increasing melt rates in the A1B scenario lead

to a reduction of thickness (only) in the coupled model, reducing its function as a dynamic barrier. Second, reducing ice draft close to the grounding lines leads to an increasing water column thickness below still deep-drafted ice, even if the grounding line position remains unchanged or grounding line migration is very small. In several locations that have already been under the floating ice shelf in the present-day situation, up to 90% of the water column thickness found at the end of the *RAnGO A1B* simulation are due to ice-shelf thinning (Fig. 10, right panel). Even though the water mass properties change only very

little between *FESOM A1B* and *RAnGO A1B* (Fig. 11), the increased water column thickness in the coupled model allows for a transport of warm water to the deepest parts of the cavity more easily. This is most notable at the estuaries of the Recovery and Support Force Glaciers, but also to the north of Bailey Ice Stream. At Support Force Glacier, the increasing melt rates in the A1B scenario lead to a reduced ice-shelf thickness and an increased slope of the ice-shelf base directly off the grounding line (Fig. 11). The latter causes 2195 annual mean along-slope ocean current velocities at the ice-shelf base to increase from about

5 cm/s in *FESOM A1B* to a maximum of 15 cm/s in *RAnGO A1B*, reinforcing stronger melt rates in this area and thus forming a positive feedback loop. Again, some of this increased melting is compensated by a reduced melting in adjacent areas, but the

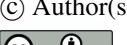



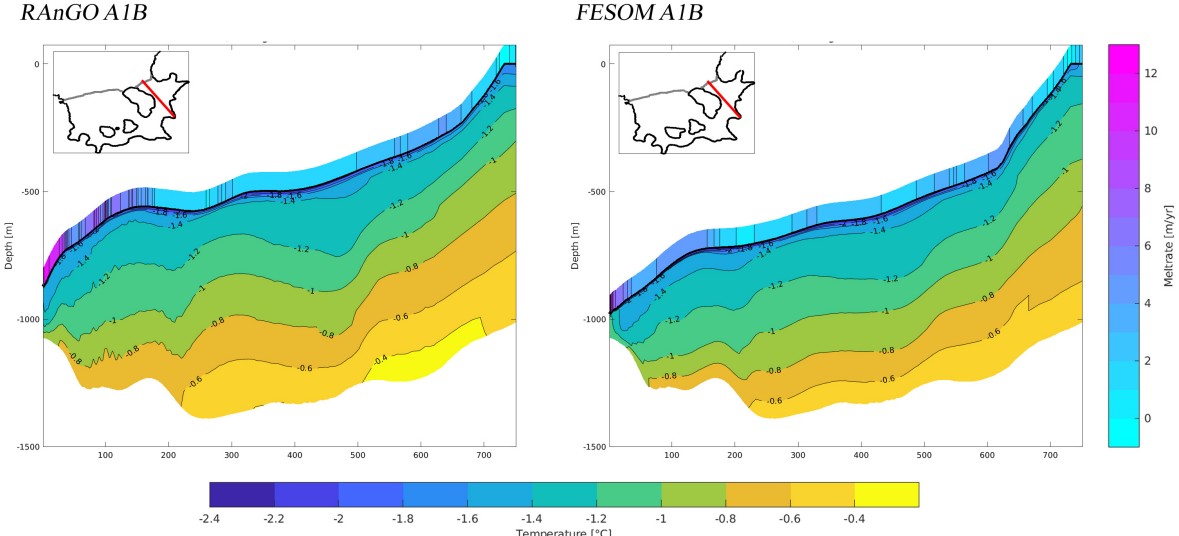

**Figure 11.** Annual mean potential temperature sections for 2195 below Filchner-Ronne Ice Shelf in *RAnGO A1B* (left) and *FESOM A1B* (right). Colors on top of the ice-shelf base indicate annual mean basal melt rates. The red lines in the maps indicate the location of the section.

residual remains positive, so that together with the increased melting along the ice front, total ice-shelf basal mass loss in the coupled model increases slightly more than in the fixed-geometry case.

### 3.4 Lessons from initial adjustment

A striking feature in the time series of Fig. 6 is the sudden reduction of FRIS mass and area from 1950 to 1951, i.e. with the

5 first coupling step. What appears as a big discontinuity merely represents 2.7 % of the original ice-shelf mass and 1.0 % of ice-shelf area. Nevertheless, this event deserves a closer look.

    While 1950 is the last year in which RIMBAY was run with parameterized melt rates, 1951 is the first year of RAnGO, i.e. the first year in which RIMBAY is forced with basal melt rates from FESOM. The top left panel in Fig. 12 thus shows the FRIS thickness distribution at the end of the 1000-yr RIMBAY spinup with Beckmann and Goose (2003) melt rates, representing the

10 *RIMBAY present-day* geometry introduced above. Using this ice thickness distribution, FESOM has been integrated with a fixed cavity geometry for 21 years. Annual-mean melt rates from the last year of this simulation (Fig. 12, bottom left panel) are fed back to RIMBAY as part of the first communication step of the coupled model. Ice thickness distribution after this first *RAnGO* year (Fig. 12, top right panel) shows that compared to the *RIMBAY present-day* geometry, ice thickness has reduced mainly in the area south of Berkner Island, i.e. between the Support Force Glacier and the Foundation Ice Stream. The time series of

15 floating ice area (Fig. 6, bottom panel) indicates that some previously floating ice has now become grounded, but grounding line migration is very small. FESOM basal melt rates obtained with this updated ice draft distribution (Fig. 12, bottom right panel) differ only very little from the result of the previous year. We conclude that variations in basal melting affect the ice-shelf



**Figure 12.** Top: Simulated ice-shelf draft for 1950 and 1951. Middle: Simulated basal melt rates averaged over the same years. Bottom: Time series of FRIS mass with green circles indicating the relevant points in time. The area covered in colors represents the modelled ice-shelf area; black lines indicate coastlines derived from RTopo-2.

thickness distribution quickly and substantially, while the feedback from a perturbed ice thickness distribution on simulated basal melt rates is much weaker.





## 4 Discussion and Conclusions

We have presented the coupled ice sheet—ice shelf—ocean model RAnGO which is focussed on the Filchner-Ronne Ice Shelf (FRIS) and the grounded ice in its catchment basin. As the reference simulation, we used a coupled model run forced with A1B scenario data from the HadCM3 climate model. Similar to the experiments of Hellmer et al. (2012) / Timmermann and

Hellmer (2013), a substantial increase of FRIS basal melt rates during the 21st/22nd centuries occurs as a response to inflowing Modified Warm Deep Water in this simulation. This event does not occur in two control simulations (coupled/uncoupled) with a perpetual 20th-century forcing from the same climate model and can thus clearly be attributed to the climate scenario/forcing data used.

Basal mass loss in the coupled A1B simulation increases by a factor of six between the simulated 1990s and the projected

2190s; maximum melt rates near the grounding line increase from 4 to 15 m/yr. Increasing melt rates lead to a thinning of the ice shelf between the present-day situation and the end of the 22nd century, especially in the deepest parts along (but not directly at) the grounding line. Maximum thickness loss in the coupled model is 280 m and occurs near the grounding line of Support Force Glacier. Grounding line migration does not exceed a distance of about 20 km. A more detailed discussion of dynamics in the various ice streams is provided in Part II of this paper.

Results from the RAnGO coupled model runs indicate that the effect of variations in ice-shelf basal melting on the ice-shelf thickness distribution is much stronger than the feedback from a perturbed ice thickness distribution on simulated basal melt rates. This is true for the rapid transition caused by switching from Beckmann and Goosse (2003) melt rates to FESOM melt rates as a boundary condition for RIMBAY at the end of the ice model spin-up; it is also true for the strongly increased melt rates projected for the end of the 22nd century, which are very similar for the *RAnGO A1B* and the *FESOM A1B* cavity

geometries despite an ice thickness difference of up to 280 m (i.e. almost 25 %). We conclude that parameterizing ice-shelf basal melt rates as a function of only (or mainly) ice-shelf thickness is not necessarily a good choice.

Although the basal melt rates are not indentical between the coupled and uncoupled simulations, our results indicate that on a time scale of up to two centuries, many aspects of ice shelf—ocean interaction at Filchner-Ronne Ice Shelf can be adressed with a fixed ice-shelf geometry even in a changing climate. The long-term trend with basal mass loss increasing by roughly a

factor of six in the A1B scenario is fully consistent between the coupled and uncoupled simulations. Year-to-year variability of basal mass loss is very similar between the coupled and uncoupled simulations both for A1B scenario forcing and in the 20th-century control runs.

A more quantitative comparison between the *RAnGO A1B* experiment and the control simulation with A1B forcing but fixed cavity geometry (*"FESOM A1B"*) reveals that the increase of basal melt rate as a response to ice-shelf cavity warming

is enhanced by about 12 % in the coupled simulation. The reduced melt potential due to the rising freezing point in areas with decreasing ice thickness in the coupled simulation is clearly outweighed by the increasing water column thickness and the increasing ice base slope, both of which cause a more efficient heat transfer and thus higher melt rates in the deepest part of the cavity. We conclude that using a fixed-geometry ice shelf—ocean model tends to attenuate rather than exaggerate the response of ice-shelf basal melt rates to ocean climate warming. The longterm evolution of ice-ocean interaction at the shores



of Antarctica under progressing climate warming and thus the projection of Antarctica's contribution to future global sea level rise, however, clearly demand for an appropriate consideration of coupled processes in regional and global climate models.

*Code availability.* Codes for RIMBAY and FESOM are available from the authors upon request.

*Author contributions.* RT set up FESOM, developed and implemented the RAnGO coupling scheme, conducted the experiments, and pre-
5    pared most of the manuscript. SG set up the RIMBAY model, provided the RIMBAY interface to the coupling routines and contributed to the preparation of the manuscript.

*Competing interests.* The authors declare that they have no conflict of interest.

*Acknowledgements.* We would like to thank H.H. Hellmer, K. Grosfeld, D. Sidorenko, and Q. Wang for helpful discussions, S. Löschke for supplying the illustration of Weddell Sea continental shelf processes, and W. Cohrs, N. Rakowsky, M. Thoma and C. Wübber for maintaining
10    excellent computing facilities at AWI. F. Schnaase created the scripts for combining sub-ice temperature sections with basal melt rate profiles. We thank the North-German Supercomputing Alliance (HLRN) for providing the computer resources required to run FESOM and RAnGO over several hundreds of years. Funding by the Helmholtz Climate Initiative REKLIM (Regional Climate Change), a joint research project of the Helmholtz Association of German research centres (HGF), has been indispensable for this study and is gratefully acknowledged.



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
