# Peer review of "Response to Filchner-Ronne Ice Shelf cavity warming in a coupled ocean—ice sheet model. Part I: The ocean perspective"

_Ocean Science, 2017_

## Referee Comment (RC1) · Anonymous Referee #1 · 6 Jun 2017

**Review of**

Response to Filchner-Ronne Ice Shelf cavity warming in a coupled ocean - ice sheet model. Part I: The ocean perspective

by Timmermann and Goeller

Summary and recommendation

This is an original paper describing the effect on ice shelf basal melt rates of coupling a dynamical ice sheet/shelf model with an ocean/sea ice model. The difference with

a fixed-geometry run is relatively small, with a small positive increase is allowance is made for dynamical adjustments. The physical processes responsible for the changes are described. The paper is well written and concise; the figures are of good quality. I suggest only minor revisions.

**Major comments**

p. 5, I. 2: "This ice model spin-up is forced by present-day surface temperatures (Comiso, 2000), accumulation rates (Arthern et al., 2006) and geothermal heat flux (Shapiro and Ritzwoller, 2004)." All these forcing datasets are somewhat outdated; please motivate your choice and discuss the impact it has on the simulated ice sheet.

p. 9, l. 4: It is not convenient to refer to figures that appear later in the paper; rather, include a map with all relevant names as Fig. 1 (or simply rename Fig. 9 to Fig. 1).

Fig. 6: Would it not be more logical if you comment on the slight drift that occurs in the RAnGO 20C control run (blue line) here, rather than later on page 14?

p. 11, l. 18: "In contrast to the former experiments, a water column thickness of only 120 m (90 m) southwest of Henry Ice Rise prevents the warm water from flushing even larger parts of the Ronne cavity in the RAnGO 20 (FESOM) simulations discussed here." Is the water column thickness so different from these former experiments, and if so, what causes this? What does this imply for the conclusions drawn in these studies when it comes to stability of the ice shelf?

Minor (textual) comments

p. 1, l. 15: Mass flux -> the Mass flux

- p. 1, l. 17: any other process -> the other processes
- p. 1, l. 19: grounding location -> grounding line location
- p. 1, l. 19: grounded ice -> grounded ice above floatation

- p. 3, l. 7: in a ramp-like shape -> by a ramp-like shape
- p. 6, l. 9: as good as possible -> as well as possible
- p. 8, l. 10: time are spent -> time is spent

p. 11, l. 8: With the beginning of the 21st century, but most notably after 2050, -> Most notably after 2050,

- p. 11, l. 12: which corresponds to a factor of six -> a factor of six increase
- p. 18, I. 22: indentical -> identical

СЗ

---

## Referee Comment (RC2) · Anonymous Referee #2 · 16 Jun 2017

This manuscript by R.Timmermann and S. Goeller describes a configuration of a coupled ice sheet-ice shelf-ocean model to investigate the impact of A1B warming scenario on the Filchner-Ronne Ice Shelf. Given the enormous impact on human society that Antarctic mass loss might have in the future this topic is highly relevant. The model is at the state of the art, the experiments described are well designed and the analysis of the results is clear. Therefore I have only minor comments to help improve the manuscript before publication.

Specific comments:

There is no comparison between the model and temperature observations, is it because there are no observations in this region or because such a comparison has already been done with previous versions of the FESOM? I imagine it is a big challenge to have a realistic ice shelf thickness and position of the grounding line because it might be highly sensitive to mean biases of ocean temperature and most ocean models have up to a few degrees of ocean temperature bias. What is the reason here for the good position of the grounding line? Is is because RAnGO has very little temperature bias or because there is room to tune the melt rate relations or because temperature is not so important in the melt rate or grounding line position change have a long time scale? It would be interesting to discuss this point in the manuscript.

How realistic is the little increase of ice shelf area during the A1B scenario? One could expect that calving would also increase and the area of ice shelf could reduce. How is calving modelled and how dependent on ice shelf thickness is it?

Also there is no comments about the feedbacks with the ice sheet. I understand this will be discussed in detail in a following paper but it would be nice to say a word also here. The thickness change in the ice shelf depend both on ocean melt rate and on ice inflow from the ice sheet, there are probably feedbacks between these because it is said in the manuscript that an increased vertical slope of the boundary between ocean and ice shelf increases ocean currents along the ice shelf which increases the melt rate.

How does the discussion of section 3.3 relate to the concept of Marine Ice Sheet Instability? Is there locations of reverse slope bed in this region? I expect that the conclusion of this study that to first order using a model with constant ice shelf topography is fine, would have been very different if the ice sheet had been on a reverse slope bed. A discussion of this point in the discussion and a comparison with other ice shelves would help put the results of this manuscript in context.

The ocean part of the RAnGO model is only in contact with an ice sheet model in the

Filchner-Ronne region, how is the interface between the ocean and the ice shelves modelled in ocean regions? In an A1B scenario one would expect that freshwater would be entering the ocean in other regions, could this have an effect on the local ocean circulation?

Small comments:

p.3, l.4: "Is" should be "it"

p.4, section 2.4, the basal melt rates are averaged yearly, is there no seasonal cycle and would it influence the ice model?.

p.8, section 2.6, why does it take so much time to build the 3D grid? The surface grid is already computed, which steps are left? In a typical free surface ocean model with sigma coordinate like ROMS, the sigma coordinates adjust vertically at each time step, this is a very fast process.

p.9, l.5-6 why is figure 7 referenced here? It does not show the in situ freezing temperature.

p.11, l.10 "my" should be "by"

p.18, l.20-21, if some work does this parameterisation it would be helpful to reference it here.

---

## Author Comment (AC1) · 10 Aug 2017

**Authors' comments on the reviews of "Response to Filchner-Ronne Ice Shelf cavity warming in a coupled ocean—ice sheet model. Part I: The ocean perspective"**

R. Timmermann and S. Goeller

August 10, 2017

First of all, we would like to thank both reviewers for their careful reading of the manuscript and their helpful and constructive comments. Your input is really appreciated.

In the following, we quote the reviewers' comments in *italic* typesetting, followed by our replies. New text added to the manuscript or modified from the original manuscript appears in **bold**.

**Anonymous Referee #2**

**Specific comments**

*There is no comparison between the model and temperature observations, is it because there are no observations in this region or because such a comparison has already been done with previous versions of the FESOM? I imagine it is a big challenge to have a realistic ice shelf thickness and position of the grounding line because it might be highly sensitive to mean biases of ocean temperature and most ocean models have up to a few degrees of ocean temperature bias. What is the reason here for the good position of the grounding line? Is is because RAnGO has very little temperature bias or because there is room to tune the melt rate relations or because temperature is not so important in the melt rate or grounding line position change have a long time scale? It would be interesting to discuss this point in the manuscript.*

Data are indeed scarce for the Weddell Sea continental shelf. CTD sections close to the ice shelf front (see, e.g., Makinson and Nicholls, JGR 1999) indicate that most of the water column is at a temperature near the surface freezing point (i.e. between $-1.8$ and $-1.9°$C), with a few warm (up to $-1.4°$C in an episodic inflow of MWDW west of Berkner Bank) and cold (down to $-2.2°$C in the ice shelf water plume on the western flank of Filchner Trough) exceptions. For an ocean model coupled to a sea-ice model, this is a comparatively easy situation: As long as the atmosphere is cold enough, the sea ice model together with a parameterization of convection will make sure that the bulk of subsurface water masses on the continental shelf will not deviate far from the surface freezing point temperature. The typical temperature bias for subsurface water masses in present-day climate FESOM and RAnGO simulations for the region discussed here does therefore not exceed 0.2 K.

We added the statement **For present-day climate, the model yields ice shelf basal melt rates, ice thickness, and grounding line location in good agreement with observations.** as the second sentence in Discussion and Conclusions.

*How realistic is the little increase of ice shelf area during the A1B scenario? One could expect that calving would also increase and the area of ice shelf could reduce. How is calving modelled and how dependent on ice shelf thickness is it?*

To some extent, the moderate grounding line retreat in RAnGO is clearly a

matter of time scales. Considering that the warm water inflow starts towards the end of the 21st century in the RAnGO A1B simulation, there is little more than one century for the ice sheet to respond to the changing situation. First response is a thinning of the ice shelf, then with some delay the thinning of the grounded ice starts. The magnitude of the delay will be shown and discussed in Part II of this paper.

Note that previous studies modeling the impact of a warm water inflow underneath the FRIS with a one-way (ocean-to-ice) coupling have found a similar increase of ice shelf area (Mengel et al., Nature Climate Change 2016), so we are not 'out of bounds' here.

Similar to many ice models, calving in RIMBAY is implemented in the following way: If the ice thickness at the ice shelf front drops below a predefined threshold (100 m in our model runs), the ice shelf at this location is considered to have become unstable and have calved. Consequently, the ice will be removed and the grid cell assumed to be open ocean. At all other places, we keep the position of the ice shelf front constant and regard the mass flux trough this front as a continuous calving process. In our model runs, ice shelf thickness never drops below the 100-m threshold, so that the calving-event criterium never applies.

*Also there is no comments about the feedbacks with the ice sheet. I understand this will be discussed in detail in a following paper but it would be nice to say a word also here. The thickness change in the ice shelf depend both on ocean melt rate and on ice inflow from the ice sheet, there are probably feedbacks between these because it is said in the manuscript that an increased vertical slope of the boundary between ocean and ice shelf increases ocean currents along the ice shelf which increases the melt rate.*

There are many ice streams feeding into the Filchner-Ronne Ice Shelf and they all have very different properties (ice velocity, ice thickness, bed slope gradient underneath the ice stream, lateral existence of buttressing mountain ranges, accumulation, proportion of ice mass above floatation where grounded, basal melt rates where floating), so it would be an oversimplification to discuss the feedbacks as a side note or in a 'rule of thumb' way - and more than that would go far beyond the ocean focus of this manuscript. In Part II of this paper, we compare all the relevant ice streams by their properties and discuss why only the Institute and the Support-Force Ice Stream show a thinning of the grounded ice and others do not.

*How does the discussion of section 3.3 relate to the concept of Marine Ice Sheet Instability? Is there locations of reverse slope bed in this region? I expect that the conclusion of this study that to first order using a model with constant ice shelf topography is fine, would have been very different if the ice sheet had been on a reverse slope bed. A discussion of this point in the discussion and a comparison with other ice shelves would help put the results of this manuscript in context.*

This as well is something we promise to adress in Part II of the paper.

*The ocean part of the RAnGO model is only in contact with an ice sheet model in the Filchner-Ronne region, how is the interface between the ocean and the ice shelves modelled in ocean regions? In an A1B scenario one would expect that freshwater would be entering the ocean in other regions, could this have an effect on the local ocean circulation?*

All other ice shelves in the coupled model are treated in the classical way with fixed geometry, so their meltwater flux responds to changes in ocean climate the same way as in fully uncoupled FESOM. Weddell Sea inflow in our setup has very similar properties between the FESOM and RAnGO simulations, so that differences in FRIS melt rates can be safely attributed to the representation of ice dynamics in this sector.

Timmermann and Hellmer (2013) discussed the effect of increased meltwater flux from FRIS on basal melt rates for Larsen C ice shelf as an example of nonlocal interaction between different ice shelves. In principal, the same mechanism is active in RAnGO, but the difference in FRIS melt rates between FESOM and RAnGO is not big enough to affect the evolution of Larsen C melt rates in any significant way.

We added the statement **All other ice shelves are modelled with fixed geometry.** towards the end of Section 2.1 (RAnGO: Overview).

**Small comments**

*p.3, l.4: "Is" should be "it"*

Done.

*p.4, section 2.4, the basal melt rates are averaged yearly, is there no seasonal cycle and would it influence the ice model?.*

A substantial seasonal cycle of basal melt rates is evident only close to the ice front, where some increase in melting is driven by summer warming of surface water. This, however, is far away from the grouding line.

Furthermore, the ice velocities at the transition between grounded and floating ice within the ice streams are about 500 m/yr on average. At the Filchner Ice Shelf front, ice velocities reach a maximum of about 1500 m/yr. If we assume the seasonal cycle of basal melting at any location to be (first order) sinusoidal, we obtain a horizontal ice thickness variation with a wave length of 500 m at the ice streams and 1500 m at the Filchner Ice Shelf front. Thus, these fluctuations are about one order of magnitude smaller than our horizontal model resolution. The application of a yearly averaged melt rate is therefore clearly justified.

*p.8, section 2.6, why does it take so much time to build the 3D grid? The surface grid is already computed, which steps are left? In a typical free surface ocean model with sigma coordinate like ROMS, the sigma coordinates adjust vertically at each time step, this is a very fast process.*

Indeed, if the variation is only in ice sheft draft or water column thickness, i.e. in adjusting the depths of coordinate levels, that's very fast. The time-consuming step is necessary (only, but in fact every time) if the surface mesh changes. We have a precomputed mesh that is adjusted to the actual cavity geometry by cutting off unneeded elements after each coupling step. This is a fast process as well, but it is only applied to the surface (triangular) grid. The time-consuming bit is the creation of the threedimensional (tehtrahedral) grid under that new surface mesh, because it does not correct the existing tetrahedral grid but creates a new 3D mesh even if only one surface element has been added or removed. We now have a more efficient machinery for this step in the pipeline though.

*p.9, l.5-6 why is figure 7 referenced here? It does not show the in situ freezing temperature.*

The citation did not refer to he freezing point but to the statement of the full sentence. We now reference the two panels (melt rates and ice draft) separately.

*p.11, l.10 "my" should be "by"*

Done.

*p.18, l.20-21, if some work does this parameterisation it would be helpful to reference it here.*

The classical case here is the parameterization of Beckmann and Gooose (2003), used by, e.g., Mengel et al. (2016), and many others. The sentence now reads **We conclude that parameterizing ice-shelf basal melt rates as a function of ice thickness, like in the widely used scheme suggested by Beckmann and Goosse (2003), is not necessarily a good appromixation to the governing processes.** This also acknowledges the fact that BG03 is very convenient for long-term ice model spinups and may well be an appropriate choice there despite its limitations.

---

## Author Comment (AC2) · 10 Aug 2017

**Authors' comments on the reviews of "Response to Filchner-Ronne Ice Shelf cavity warming in a coupled ocean—ice sheet model. Part I: The ocean perspective"**

R. Timmermann and S. Goeller

August 10, 2017

First of all, we would like to thank both reviewers for their careful reading of the manuscript and their helpful and constructive comments. Your input is really appreciated.

In the following, we quote the reviewers' comments in *italic* typesetting, followed by our replies. New text added to the manuscript or modified from the original manuscript appears in **bold**.

**Anonymous Referee #1**

**Major comments**

*p. 5, l. 2: "This ice model spin-up is forced by present-day surface temperatures (Comiso, 2000), accumulation rates (Arthern et al., 2006) and geothermal heat flux (Shapiro and Ritzwoller, 2004)." All these forcing datasets are somewhat outdated; please motivate your choice and discuss the impact it has on the simulated ice sheet.*

It is true that these are not the most recent data sets, but to our know knowledge they are still the ones used by the most ice modelers. Like with any boundary condition and any parameterization, the choice of data sets and the choice of parameters are not independent steps - in fact they are so closely connected that parameter values and data sets can righteously be treated as married pairs. Given that RIMBAY has been comprehensively tuned to today's observations with the abovementioned data sets (Sutter et al., GRL 2016) and gives quite acceptable results in terms of ice shelf thickness and grounding line locations in the configuration used for RAnGO, we are confident that we are not introducing critical errors here.

*p. 9, l. 4: It is not convenient to refer to figures that appear later in the paper; rather, include a map with all relevant names as Fig. 1 (or simply rename Fig. 9 to Fig. 1).*

Given that the names actually occur in the dicsussion of Fig. 7, we will add acronyms **SF (Support Force Glacier), FI (Foundation Ice Stream), EI (Evans Ice Stream), RI (Rutford Ice Stream), H (Henry Ice Rise), K (Korff Ice Rise), RT (Ronne Trough) and FT (Filchner Trough)** to all panels of Fig. 7. The caption will be adjusted accordingly. The long list of acronyms is clearly not ideal, but we found that the panels are too small to allow for full-text labels. At least all the information is available to the reader without having to refer to some other figure now.

*Fig. 6: Would it not be more logical if you comment on the slight drift that occurs in the RAnGO 20C control run (blue line) here, rather than later on page 14?*

The discussion on page 14 is on ice shelf thickness, area and mass (Section 3.2), while the first appearance and discussion of Fig. 6 is in the Basal melt rates and hydrography section (3.1). Section 3.1.3 actually addresses melt rates in the FESOM and RAnGO 20C control experiments, but we agree that the structure is not perfectly clear here. So, we changed the relevant section title to **3.1.3 FESOM and RAnGO 20C control experiments** to have naming consistent with the scenario section 3.1.2. Furthermore, section 3.1.1 has been renamed to **3.1.1 Present-day climate in FESOM and RAnGO** to make the distinction between 'hindcast' and projection more visible.

*p. 11, l. 18: "In contrast to the former experiments, a water column thickness of only 120 m (90 m) southwest of Henry Ice Rise prevents the warm water from flushing even larger parts of the Ronne cavity in the RAnGO 20 (FESOM) simulations discussed here." Is the water column thickness so different from these former experiments, and if so, what causes this? What does this imply for the conclusions drawn in these studies when it comes to stability of the ice shelf?*

We added further detail to the statement. The sentence now reads: **In contrast to the former FESOM experiments, which adopted a water column thickness of about 200 m southwest of Henry Ice Rise from RTopo-1 (Timmermann et al., 2010), slightly thicker ice in RIMBAY and a better representation of bottom topography in the RAnGO (FESOM) simulations discussed here lead to a water column thickness of only 120 m (90 m) in the channel and thus prevent a rapid spreading of warm water into the Ronne cavity.**
     With increasing melt rates, the ice southwest of Henry Ice Rise gradually becomes thinner in RAnGO, so that warm water can pass the channel slightly more easily. The difference in bottom temperatures in the Ronne cavity between the two simulations is not very big though and the melt rate difference in Fig. 10 (left panel) does not show any signature of this process, so we decided to not discuss it in the paper.

**Minor (textual) comments**

*p. 1, l. 15: Mass flux $->$ The mass flux*

Done.

*p. 1, l. 17: any other process $->$ the other processes*

We decided to go with **all other proceses**.

*p. 1, l. 19: grounding location − > grounding line location*

Done

*grounded ice − > grounded ice above floatation*

Good point. We rephrased the sentence into **Changes in ice-shelf thickness and grounding line location may therefore alter the discharge of ice grounded above floatation and thus contribute to global sea level rise.**

*p. 3, l. 7: in a ramp-like shape − > by a ramp-like shape*

Done.

*p. 6, l. 9: as good as possible − > as well as possible*

Done.

*p. 8, l. 10: time are spent − > time is spent*

Done.

*p. 11, l. 8: With the beginning of the 21st century, but most notably after 2050,
− > Most notably after 2050,*

Done.

*p. 11, l. 12: which corresponds to a factor of six − > a factor of six increase*

Done.

*p. 18, l. 22: indentical − > identical*

Done.